# Using Blood Eosinophil Count as a Biomarker to Guide Corticosteroid Treatment for Chronic Obstructive Pulmonary Disease

**DOI:** 10.3390/diagnostics11020236

**Published:** 2021-02-03

**Authors:** Pradeesh Sivapalan, András Bikov, Jens-Ulrik Jensen

**Affiliations:** 1Department of Internal Medicine, Respiratory Medicine Section, Herlev-Gentofte Hospital, 2900 Hellerup, Denmark; jens.ulrik.jensen@regionh.dk; 2Department of Internal Medicine, Zealand University Hospital, 4000 Roskilde, Denmark; 3Wythenshawe Hospital, Manchester University NHS Foundation Trust, Manchester M23 9LT, UK; andras.bikov@gmail.com; 4Division of Infection, Immunity and Respiratory Medicine, University of Manchester, Manchester M13 9NT, UK; 5Department of Clinical Medicine, Faculty of Health Sciences, University of Copenhagen, 2200 Copenhagen, Denmark

**Keywords:** chronic obstructive pulmonary disease, blood eosinophil count, systemic corticosteroids, biomarkers, inhaled corticosteroids

## Abstract

Treating patients hospitalised with acute exacerbations of chronic obstructive pulmonary disease (COPD) usually involves administering systemic corticosteroids. The many unwanted side effects associated with this treatment have led to increased interest in minimising the accumulated corticosteroid dose necessary to treat exacerbations. Studies have shown that short-term treatment with corticosteroids is preferred, and recent trials have shown that biomarkers can be used to further reduce exposure to corticosteroids. Interestingly, high eosinophil counts in patients with acute exacerbations of COPD are indicative of an eosinophilic phenotype with a distinct response to treatment with corticosteroids. In addition, post-hoc analysis of randomised control trials have shown that higher blood eosinophil counts at the start of the study predict a greater response to inhaled corticosteroids in stable COPD. In this review, we examine the studies on this topic, describe how blood eosinophil cell count may be used as a biomarker to guide treatment with corticosteroids, and identify some relevant challenges.

## 1. Introduction

Personalised treatment and precision medicine have recently attracted much interest due to their potential benefits, which include decreasing the administration of unnecessary treatments and identifying patients who may benefit most from specific medications. A personalised approach to treatment may be guided by genetic information, monitoring relevant clinical events or using biomarkers [1].

Chronic obstructive pulmonary disease (COPD) is a common disorder which is characterised by progressive airflow limitation due to noxious particles, such as cigarette smoke [2]. These particles induce inflammation in the lung which may lead to tissue destruction and remodelling, excessive mucus secretion and impaired anti-microbial response [3]. However, the extent and the nature of airway inflammation in patients with COPD is variable with a proportion of patients having high eosinophil granulocyte levels in the airways. Eosinophil granulocytes are cells of the immune system that release anti-microbial granules and produce cytokines that play a role in the defence against parasites and participate in allergic reactions [4]. Their levels are increased in patients with inflammatory diseases such as asthma, allergic rhinitis and several skin diseases [5]. Their role in COPD is not fully clear, however patients with eosinophilia may benefit from a different treatment than those without high levels of eosinophils. A key challenge for physicians is to identify patients with a suitable benefit/risk profile for a particular treatment. Corticosteroids are anti-inflammatory drugs that are indicated in both stable COPD and at exacerbation [2]. However, due to their potential short and long-term side effects, a simple biomarker associated with a beneficial treatment response to corticosteroids in patients with COPD would be extremely valuable. This review focuses on the eosinophilic COPD phenotype, and whether using blood eosinophil levels as a biomarker may decrease the exposure to corticosteroids in patients with COPD.

## 2. Decreasing Exposure to Corticosteroid Treatment

Acute exacerbations of chronic obstructive pulmonary disease (AECOPD) are among the most frequent reasons for hospital admission globally, and these can be fatal [6]. When patients with COPD experience exacerbations that require hospitalisation, the initial standard treatment includes the administration of systemic corticosteroids for 5–7 days [7,8] as well as bronchodilator treatment. Corticosteroids reduce inflammation, decrease the risk of treatment failure and promote recovery from symptoms [9]. However, potential side effects are associated with corticosteroid treatment, including increased risk of infections, osteoporosis, adrenal insufficiency, venous thromboembolisms and hyperglycaemia [10,11,12,13,14,15,16]. Therefore, minimising the accumulated dose of corticosteroids while effectively treating exacerbations is very important. Previous research has shown that blood eosinophil levels may be used as a biomarker to decrease the corticosteroid dose administered to patients with COPD [17].

In 2014, a meta-analysis by Walters et al. showed that treating AECOPD with corticosteroids increased the forced expiratory volume in 1 s (FEV_1_) during the first 72 h with approximately 100 mL as compared to not treating with these drugs and reduced the risk of treatment failure, relapse and length of hospital stay, compared with placebo [18,19]. Further studies have investigated decreasing the 14-day systemic corticosteroid treatment period. In 2013, the REDUCE trial [20] investigated whether a 5-day treatment was non-inferior to the standard treatment at that time, 14 days of systemic corticosteroid treatment. The study found that both time until a subsequent exacerbation and death within six months did not differ significantly between the two groups, suggesting that the short lower-dose treatment was non-inferior [8]. Following these results, the Global Initiative for Chronic Obstructive Lung Disease (GOLD) recommended that systemic corticosteroids should be administered to patients with AECOPD for only 5–7 days [7,8]. An observational cohort study from 2019 compared the risk of pneumonia hospitalisation and all-cause mortality within one year between patients who were given short- and long-term oral corticosteroid treatments [21]. The study concluded that both risks increased in patients who were given the longer-term treatments.

## 3. COPD Eosinophil Phenotypes

Eosinophils are pleiotropic leukocytes and major effector cells of the immune system. Approximately 0.5–1% of the white blood cells in healthy individuals are eosinophils. How eosinophils infiltrate the airways of patients with COPD is not completely understood [22]. Eosinophils are derived from the bone marrow and circulate briefly in the blood before being distributed to locations such as the gastrointestinal tract and thymus [22,23]. However, they are not generally found in the lung, and their presence in that organ is indicative of an abnormal inflammatory reaction [24]. Elevated eosinophil levels have been detected in the airways of patients with COPD from sputum, bronchoalveolar lavage and bronchial biopsy samples, indicating that this type of inflammation is present in a subgroup of patients with COPD [25,26] and it is often associated with non-infectious exacerbations [27]. Eosinophilic airway inflammation has been detected in up to 40% of patients with COPD (based on the sputum criterion) [28,29,30,31,32], during stable periods of the disease. It has also been found in 28% of patients with exacerbations [27] and may be used to characterise specific COPD phenotypes. In patients with COPD, an increased sputum eosinophil count was associated with better lung function, more future exacerbations, and symptoms that responded better to treatment with inhaled [33] and oral corticosteroids [30,32,33].

Therefore, sputum eosinophil levels may be used to quantify airway eosinophilia, identify a particular type of COPD, and guide corticosteroid treatment for patients with AECOPD. However, evaluating patient sputum samples is technically challenging and time consuming and the results may lack reproducibility. More particularly, some patients cannot produce sputum spontaneously and induction with hypertonic saline may be necessary to obtain a sample. However, sputum induction may cause bronchoconstriction in some patients [34], therefore it is not feasible in patients with very severe airflow limitation and those with AECOPD. Since using sputum eosinophil counts would require significant additional investment of hospital resources and training of staff [35,36,37], clinicians have suggested using blood eosinophil counts as a surrogate to identify the eosinophilic COPD phenotype [38]. Blood eosinophils may be quantified routinely without additional resources, time or equipment. Furthermore, peripheral blood and sputum eosinophil levels are moderately correlated during exacerbations [27,34,39,40], as demonstrated by a cluster analysis which showed that sputum eosinophilia in patients with COPD exacerbations was predicted by a blood eosinophil count of >2% with 60% specificity and 90% sensitivity [27,41]. One of the major challenges in clinical decision making is having reliable biomarkers available at the point of care. Blood eosinophil levels may fluctuate significantly in a single individual and are influenced by factors such as medication and comorbidities. In the “Evaluation of COPD Longitudinally to Identify Predictive Surrogate Endpoints cohort” (ECLIPSE) study (a multicentre multinational 3-year observational study that enrolled 2000 patients with COPD), only 37% of patients with COPD yielded consistent blood eosinophil counts (≥2%) at all visits over three years [40]. In another study, blood eosinophil levels fluctuated above and below a threshold of 300 cells/µL over two years in 41% of participants [42], whereas only 15% of participants showed persistently high (≥300 cells/µL) blood eosinophil levels. Consequently, if treatment was to be based on blood eosinophil counts, many patients in this study would alternate between two treatment groups depending on the measurement times. Despite these variations, the overall findings support the applicability of using repeated blood eosinophil measurements to identify the eosinophilic COPD phenotype.

There are other surrogate markers to detect airway eosinophilia than blood eosinophils. Fractional exhaled nitric oxide (FENO) is a frequently used non-invasive marker of type two airway inflammation in asthma and is a tool to guide corticosteroid treatment [43]. However, its utility in COPD is limited as smoking and oxidative stress could lead to lower FENO values [44]. In line with this, FENO in COPD had little additive value compared to blood eosinophils [45]. Blood levels of eosinophilic cationic protein (ECP) could be another marker of airway eosinophilia. Indeed, higher plasma ECP levels were related to the burden of exacerbations in COPD [46]. However, other studies did not find a relationship between blood ECP concentrations and eosinophil counts in sputum [47,48], therefore, the role of ECP needs to be explored in future studies.

## 4. The Eosinophilic COPD Phenotype and Future Risk

Patients with elevated eosinophil levels during exacerbations may be at higher risk of complications, future exacerbations, pneumonia, longer hospital stays and mortality [49,50,51,52,53,54]. However, these results are inconclusive as other studies [55,56] did not find differences in readmission rates between eosinophilic and non-eosinophilic exacerbations. Several studies have reported an association between high eosinophil counts at stable disease and an increased risk of exacerbations [50,51,57]. Similarly, a study of two large long-term prospective cohorts, “The Genetic Epidemiology of COPD (COPDGene) cohort” (a multicentre observational study that enrolled more than 10,000 smokers) [58] and the ECLIPSE cohort [59], observed a linear association between absolute blood eosinophil counts at stable disease and exacerbation risk [50,60].

A recent study on patients with COPD observed that those with high blood eosinophil counts (≥340/µL) were at greater risk of hospitalisation from pneumonia. However, this study found no significant interaction between treatment with inhaled corticosteroids (ICS) and blood eosinophil counts on the risk of exacerbations [61]. In summary, these findings show that patients with an eosinophilic COPD phenotype may be at greater risk of negative outcomes. Sometimes blood eosinophils also have been shown to have a protective effect in COPD. Higher blood eosinophils have been associated with fewer infections and possibly a better survival, but the mechanisms to explain these findings have not been defined [40,62,63,64,65,66,67].

During exacerbations, patients with an eosinophilic COPD phenotype also show a better response to corticosteroid treatment than patients with neutrophilic inflammation [27,32,68]. Therefore, patients with COPD and high eosinophil counts may benefit more from corticosteroid treatment than patients with low eosinophil counts, perhaps suggesting that corticosteroids should not be administered to the latter group. Consequently, further studies are needed to investigate whether biomarkers such as blood eosinophil levels may be used to determine when to administer corticosteroids [25].

## 5. Blood Eosinophil-Guided Inhaled Corticosteroid Treatment of Patients with Stable COPD

Inhaled corticosteroids (ICS) are often administered as a maintenance treatment for patients with COPD [62]. ICS are anti-inflammatory drugs given to prevent exacerbations, and they may improve lung function when combined with other medications such as long-acting beta agonists (LABA) [8]. However, long-term ICS treatment may have adverse effects, potentially leading to osteoporosis, diabetes and cataracts [10]. Furthermore, there may be an increased risk for pneumonia in patients with COPD who use ICS. This risk however strongly depends on the corticosteroid component and the device [69]. COPD patients with predominantly eosinophilic airway inflammation [27] may gain the most benefit from ICS [31,70]. The GOLD guidelines have changed to reflect the recent finding that eosinophil levels in patients with COPD may influence the response to ICS [71]. More particularly, ICS are indicated for patients with a significant risk for exacerbation and increased blood eosinophils. Evidence for this association has been derived from a series of post-hoc analyses of randomised controlled trials (RCTs), Table 1. When patients with COPD were treated with different LABA versus LABA/ICS combinations [57,70,72,73], ICS showed the strongest protection against exacerbations in patients with high blood eosinophil counts. The IMPACT (*n* = 10,333) RCT compared triple inhalation therapy with dual therapy (either ICS–LABA or long-acting muscarinic antagonists (LAMA)–LABA) in patients with COPD and found that triple therapy decreased the risk of exacerbations significantly. This study also showed that the greatest reduction in exacerbations occurred in patients with high blood eosinophil counts, suggesting that this group responded better to treatment with ICS [74]. Studies have also investigated the effect of withdrawing ICS following a period of triple therapy [75,76,77]. A post-hoc analysis of data from the WISDOM study of ICS withdrawal in patients on triple therapy with combined LABA/ICS and long-acting muscarinic antagonists (LAMA) found that patients with higher blood eosinophil counts were more likely to develop exacerbations after withdrawal of ICS, with a significant treatment-by-subgroup interaction above an eosinophil count of ≥4% or >300 cells/µL [78]. In addition, patients with eosinophilia and previous exacerbation burden were those who benefited most from ICS continuation [75]. Another study, the randomised double-blind SUNSET trial, also investigated the withdrawal of ICS after a 4-week period of triple therapy compared to continuous triple therapy [77]. Post-hoc analysis found that patients with eosinophil cell counts >300 cells/µL had a greater risk of exacerbations when ICS were withdrawn [79]. Similarly, a post-hoc analysis [57] of the FORWARD randomised double-blind study [80] investigated how baseline blood eosinophil levels affected different endpoints such as the risk of COPD exacerbations over a 48-week period and changes in FEV_1_ pre-dose in the morning after 48 weeks of treatment with either a combination of ICS and LABA or LABA alone. The study found that the frequency of exacerbations increased with increasing eosinophil count in patients treated with LABA monotherapy, with 1.39 events/patient/year within the highest quartile (≥279.8/μL). Furthermore, combining ICS and LABA had the greatest effect on increasing FEV_1_ in patients in the highest blood eosinophil quartile. These findings favour treating patients with high eosinophil counts with a combination of ICS and LABA because these patients will benefit most from treatment supplemented with ICS. Some studies do not support using blood eosinophil levels to guide ICS treatment of patients with COPD. A post-hoc analysis of the ISOLDE study found that ICS decreased the risk of exacerbations more markedly in patients with low levels of eosinophils (<2%) and that the time to the first exacerbation did not differ between the high and low eosinophil level groups [81]. This has recently been clarified in another post-hoc analysis of the same trial reporting that changes in blood eosinophil counts following corticosteroid treatment, rather than baseline values are predictive for corticosteroid response [82]. The FLAME study found that LABA/LAMA was more effective than LABA/ICS treatment in preventing COPD exacerbations, regardless of whether the blood eosinophil count was high or low, although a cut-off level of <600 eosinophils/mL was required for study entry [83]. Thus, most studies suggest that using blood eosinophil levels to guide ICS treatment of patients with an eosinophilic COPD phenotype decreases the risk of exacerbations and improves lung function, whilst sparing patients from being exposed to unnecessary corticosteroids when these would be of little benefit. Importantly, all of these recommendations are based on post-hoc studies and no RCT has investigated eosinophil-guided corticosteroid treatment of patients with COPD [42,64,84,85]. Currently, there is one multicentre RCT attempting to validate these findings in severe–very severe COPD (COPERNICOS trial) (NCT04481555), and the results are due by 2025.

## 6. Blood Eosinophil-Guided Systemic Corticosteroid Treatment of Patients with COPD

Since discovering that decreased corticosteroid exposure is beneficial for patients and that there are different COPD phenotypes, studies have investigated whether biomarkers may be used to guide systemic corticosteroid treatment. In 2012, a single-centre double-blind RCT [17] investigated the possibility of using blood eosinophil levels to decrease corticosteroid treatment dose. Patients were randomised to receive either prednisolone for exacerbations or biomarker-directed prednisolone therapy. In the biomarker-directed therapy, patients who had exacerbations and blood eosinophil levels of >2% received prednisolone, whereas those with eosinophil levels of ≤2% did not. Both groups also received antibiotics. This non-inferiority study found no difference in treatment failure rate or health status between the groups. However, only 10 patients who were admitted to hospital due to exacerbations were included in this study. Therefore, these findings cannot be generalised to severe acute exacerbations. Furthermore, this study used a single blood eosinophil measurement, recorded at the onset of exacerbations, to determine treatment for the entire study period. Because blood eosinophil counts may fluctuate significantly in individual patients, studies in which serial blood eosinophil measurements are performed may be more informative. Similarly, a meta-analysis of three clinical trials (*n* = 243) evaluated the effectiveness of using blood eosinophil-guided systemic corticosteroids to treat AECOPD, compared with placebo. The primary outcome was treatment failure, which was 66% among patients with high blood eosinophil counts (≥2%) who did not receive corticosteroids and 11% among those who did receive corticosteroids. There was no difference between the same groups among patients with blood eosinophil counts of <2% [86].

The CORTICO-COP trial investigated whether a biomarker-guided algorithm based on serial daily blood eosinophil counts from the admission day and every morning while admitted (maximum five days) could be used to reduce the exposure to systemic corticosteroids while being non-inferior to standard treatment of AECOPD [87] regarding admission length and death (measured by “days alive and out of hospital”). A total of 318 patients were randomised into two treatment groups: one receiving corticosteroid treatment based on daily blood eosinophil counts and the other receiving standard treatment with corticosteroids for five days. For the eosinophil-guided group, corticosteroid treatment was withheld on days where eosinophil counts were <300 cells/µL, whereas patients received 37.5 mg of prednisolone on days where eosinophil counts were ≥300 cells/µL (Figure 1). There was no difference between the two groups in the primary endpoint, days alive and out of hospital within 14 days from recruitment, and the non-inferiority assumption was thus met. In addition, secondary endpoints such as worsening of diabetes, occurrence of new onset diabetes or infections requiring antibiotics were assessed from baseline to 30- and 90-day follow-up points. The results showed that the biomarker-guided treatment approach was non-inferior to standard treatment for AECOPD. Furthermore, the duration of systemic corticosteroid treatment was substantially lower for patients who received treatment based on daily blood eosinophil measurements (Figure 2). The eosinophil-guided approach led to a reduction in median corticosteroid treatment durations from five days to two days, with approximately two-thirds of patients in the eosinophil-guided group being off corticosteroids on any given day throughout the study, except for day one (Figure 2). This trial overcame the challenge of fluctuating eosinophil levels and was able to compare a biomarker-guided approach to standard treatment with corticosteroids for five days.

## 7. Challenges and Further Directions

However, a number of issues remain unresolved. Thus, the threshold level of blood eosinophils that can be used for decision-making in clinical practice has not yet been finally determined. Prospective clinical studies are needed to determine the threshold level of blood eosinophils, which can be used both to predict the effects of ICS and to predict the risk of future exacerbations. In addition, the majority of the clinical trials predicting exacerbation rates, and their reduction with ICS, examined COPD cohorts enriched for subjects with frequent exacerbations and sometimes with a history of asthma. Few data are available in non-asthmatic COPD subjects who were not frequent exacerbators, and these data did not truly support the role of blood eosinophils as predictor of COPD exacerbations [42,64,84,85]. Therefore, larger randomised controlled trials with stratification by blood eosinophil count are needed to validate these findings.

Furthermore, current smoking, oxidative stress and infections contribute to a variable degree of resistance to corticosteroids in COPD [89]. This resistance could at least partially be resolved by administering long-acting muscarinic antagonists [90], macrolides [91], roflumilast [92], carbocisteine [93] or theophylline [94], medications that are all commonly used in patients with COPD. Airway eosinophilia in COPD is not necessarily interrelated with these factors and medications, therefore these could be biases in clinical trials investigating the predictive value of blood eosinophils. On one hand, further trials should take into account these confounding factors. On the other hand, more comprehensive biomarkers than blood eosinophilia could assist in predicting response to corticosteroids. Neutrophils to lymphocytes and platelets to lymphocytes ratios could be potential aids as their combination with blood eosinophils better predicted steroid resistance than blood eosinophils alone [95].

The European Respiratory Society and the American Thoracic Society guidelines for the management of patients with AECOPD note that identifying the phenotype in which a response to corticosteroid therapy occurs is a line of research that should be continued [96].

## 8. Conclusions

There has been a steady stream of recent studies investigating the use of eosinophil levels as biomarkers in patients with COPD. Many of these studies highlight the value of using blood eosinophil counts to identify patients with particular COPD phenotypes and administering both inhaled and systemic corticosteroids to treat patients with high levels of eosinophils. Overall, this personalised approach to treatment promises to decrease patient exposure to unnecessarily high doses of corticosteroids. However, there is a need to validate these findings in well-conducted randomised trials before we can make use of this in the treatment of COPD patients.

## Figures and Tables

**Figure 1 diagnostics-11-00236-f001:**
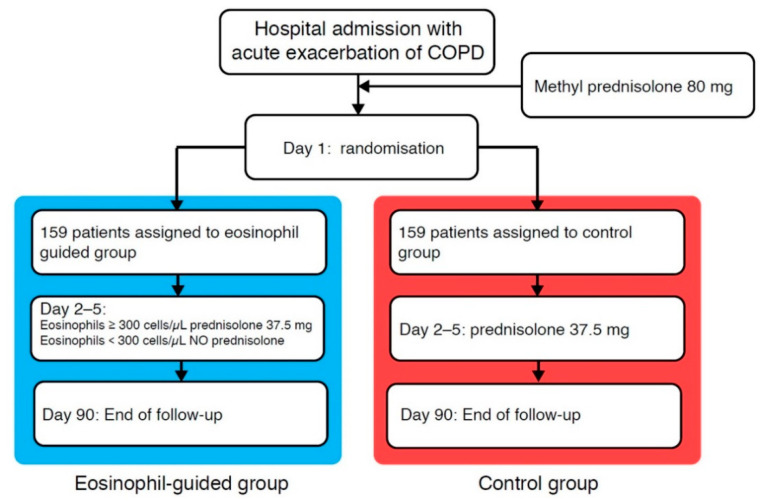
Treatment algorithm from the CORTICO-COP trial—adapted from Sivapalan et al. Abbreviation: COPD, chronic obstructive pulmonary disease.

**Figure 2 diagnostics-11-00236-f002:**
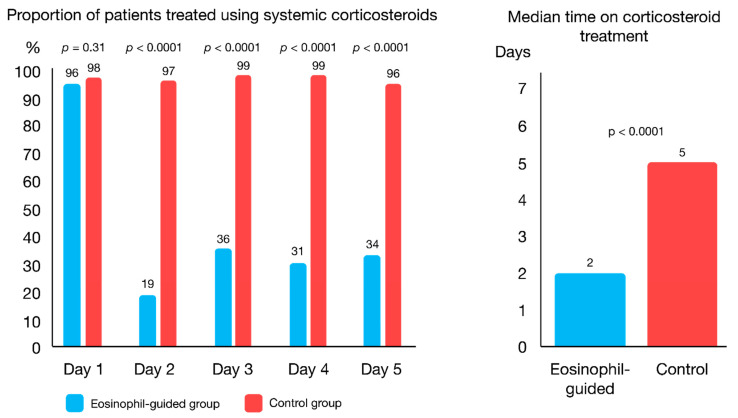
Fraction of patients on systemic corticosteroid treatment and the median treatment length in the CORTICO-COP trial—adapted from Sivapalan et al.

**Table 1 diagnostics-11-00236-t001:** Eosinophil thresholds used to identify systemic and inhaled corticosteroid–responsive COPD exacerbations in recent COPD clinical trials.

Author	*N*	Treatment	Eosinophil Threshold	Findings
IMPACT—Lipson et al., 2018 [74]	6204	Fluticasone furoate (ICS)/vilanterol (LABA) vs. umeclidinium (LAMA)/vilanterol (LABA)	≥150 cells/μL	ICS/LABA/LAMA vs. LABA/LAMA, BEC ≥ 150 cells/μL: 32% reduction in annual exacerbation rate *
Bafadhel et al., 2018 [70]	4528	BDP/FOR vs. FOR	200–340 cells/μL	26–50% decrease in annual exacerbation rate
			350–630 cells/μL	51–60% decrease in annual exacerbation rate
FORWARD—Siddiqui et al., 2015 [57]	1199	BDP/FOR vs. FOR	≥279.8 cells/μL	BEC ≥ 279.8 cells/μL: 46% reduction in annual exacerbation rate. BEC < 279.8 cells/μL: 28% reduction in annual exacerbation rate.
ISOLDE—Barnes et al., 2016 [81]	456	fluticasone propionate versus placebo	≥2%	The reduction in overall exacerbation rate for fluticasone propionate versus placebo was higher in the <2% eosinophil group compared with the ≥2% eosinophil group
FLAME—Anzueto et al., 2016 [88]	3362	Glycopyrronium (LAMA)/indacaterol (LABA) vs. fluticasone (ICS)/salmeterol (LABA)	≥2%	11% reduction (LAMA/LABA vs. ICS/LABA) **
Calverley et al., 2017 [75]	2420		≥300 cells/µL	ICS withdrawal vs. continuation ≥1 exacerbation in prior year AND BEC ≥ 300 cells/μL: 45% increase in annual exacerbation rate * ≥2 exacerbations in prior year AND BEC ≥ 300 cells/μL: 75% increase in annual exacerbation rate *;
WISDOM—Watz et al., 2016 [76]	2296	FP/SAL/tiotropium vs. continuation or withdrawal of ICS treatment	≥300 cells/µL	ICS withdrawal vs. continuation: BEC < 300 cells/μL *: 4% increase *; BEC ≥ 300 cells/μL *: 56% increase in annual exacerbation rate * BEC < 400 cells/μL: 7% increase *; BEC ≥ 400 cells/μL: 73% increase in annual exacerbation rate *
SUNSET—Chapman et al., 2018 [77]	527	Tiotropium (LAMA)/salmeterol (LABA)/fluticasone (ICS). Two arms: first group continues with triple therapy; second switches to glycopyrronium (LAMA)/indacaterol (LABA)	≥300 cells/µL	ICS withdrawal vs. continuation: BEC ≥ 300 cells/μL: 86% increase in annual exacerbation rate

***** Exacerbation rate for moderate and severe exacerbations. ** Exacerbation rate for mild, moderate and severe exacerbations. Abbreviations: AECOPD = acute exacerbations of chronic obstructive pulmonary disease, COPD = chronic obstructive pulmonary disease, BDP = budesonide dipropionate, FOR = formoterol, FP = fluticasone propionate, LABA = long-acting beta agonists, LAMA = long-acting muscarinic antagonists, SAL = salbutamol, ICS = inhaled corticosteroids, BEC: blood eosinophil count, FORWARD = Foster 48-Week Trial to Reduce Exacerbations in COPD, IMPACT = Informing the Pathway of COPD Treatment, ISOLDE = inhaled steroids in obstructive lung disease in Europe, FLAME = Fluticasone Salmeterol on COPD Exacerbations, WISDOM = Withdrawal of Inhaled Steroids during Optimized Bronchodilator Management, SUNSET = Study to Understand the Safety and Efficacy of ICS Withdrawal from Triple Therapy in COPD.

## Data Availability

Not applicable.

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
