# Peer review of "Using Blood Eosinophil Count as a Biomarker to Guide Corticosteroid Treatment for Chronic Obstructive Pulmonary Disease"

_diagnostics, 2021, doi:10.3390/diagnostics11020236_

Round 1

Reviewer 1 Report

Quite a lot of works have been devoted to the study of the prognostic value of blood eosinophils. The presented review summarizes well the results of recent studies, but the novelty is not obvious to me. The conclusion of the review further emphasizes the fact that patients with elevated eosinophil levels in COPD respond better to corticosteroid therapy. However, this is already a known fact. Nevertheless, a number of issues remain unresolved. Thus, the threshold level of blood eosinophils that can be used for decision-making in clinical practice has not yet been finally determined. Prospective clinical studies are needed to determine the threshold level of blood eosinophils, which can be used both to predict the effects of ICS and to predict the risk of future exacerbations. The European Respiratory Society and the American Thoracic Society guidelines for the management of patients with acute exacerbations of COPD note that identifying the phenotype in which a response to corticosteroid therapy occurs is a line of research that should be continued. There is also evidence that a high level of eosinophilic cationic protein in the blood plasma in patients with COPD is associated with a more severe course and development of more frequent infectious-dependent exacerbations of the disease, requiring the administration of inhaled glucocorticosteroids and antibiotics (https://doi.org/10.20538/1682- 0363-2020-1-59-66), which allows using this indicator as a predictor of the severity of COPD. It has been shown that patients with COPD resistant to corticosteroid therapy compared with sensitive patients are characterized by an increase in the level of a factor that inhibits the migration of macrophages, the ratio of the absolute numbers of neutrophils to lymphocytes and platelets to lymphocytes, and not just a decrease in the absolute and relative numbers of eosinophils (https: // doi .org / 10.18093 / 0869-0189-2018-28-6-681-692). It seems to me that the article is of no interest in its current form, since there is no new look at the problem. Perhaps other data on the use of a complete blood count should be added. Or you need to highlight the article differently.

Reviewer 2 Report

  • This review is well written and supported by key references. Nevertheless, it adds little to what has been already published because, in the last few years, a lot of papers including review and editorial had emphasized the potential role of blood eosinophil count as a biomarker for prediction of exacerbation and response to ICS treatment.
  •  
  • What is less known is that the majority of clinical trials predicting exacerbation rates, and their reduction with ICS, examined COPD cohorts enriched for subjects with frequent exacerbations and sometimes with a history of asthma. Few data are available in non-asthmatic COPD subjects who were not frequent exacerbators, and these data did not truly support the role of blood eosinophils as predictor of COPD exacerbations. This concept should be included together with the appropriate references (Am J Respir Crit Care Med 2018, 197: 1216-1218; ERJ  Open Research 2018, 4:00022-2018; Int J Chronic Obstr Pulm Dis 2017, 12:1819-1824; J Clin Med 2019, 8:962 and reference 43 of the current review).
  •  
  • Eosinophils have both deleterious and protective effects on COPD, including reduced mortality. In paragraph 4th the authors underline that patients with an eosinophilic phenotype may be at greatest risk of negative outcomes. A discussion of the protective effects of eosinophils should also be included (Eur Respir J 2014, 44:1697-1700; Respir Med 2016, 112:88-96; Lancet Respir Med 2016, 4:731-741).
  •  
  • 3rd paragraph, page 3 lines 101-102: “Sputum eosinophils are moderately correlated with airway inflammation (27, 35, 36)” All these references examined the relation between blood and sputum eosinophils, none of them included data on lung inflammation.
  •  
  • Figure 2: unit of measure should be include in the y axis

Round 2

Reviewer 1 Report

The authors made all the corrections in accordance with the recommendations of the reviewer. In its present form, the article can be recommended for publication.

Reviewer 2 Report

The changes done by the authors to the manuscript are satisfactory